# Electronic photoreceptors enable prosthetic visual acuity matching the natural resolution in rats

Bing-Yi Wang [1,7] ✉, Zhijie Charles Chen [2,7] ✉, Mohajeet Bhuckory[3,4], Tiffany Huang[2], Andrew Shin[5], Valentina Zuckerman[3], Elton Ho [3,4], Ethan Rosenfeld [3], Ludwig Galambos[2], Theodore Kamins [2,3], Keith Mathieson [6] & Daniel Palanker [3,4]

Localized stimulation of the inner retinal neurons for high-acuity prosthetic vision requires small pixels and minimal crosstalk from the neighboring electrodes. Local return electrodes within each pixel limit the crosstalk, but they over-constrain the electric field, thus precluding the efficient stimulation with subretinal pixels smaller than 55 µm. Here we demonstrate a high-resolution prosthetic vision based on a novel design of a photovoltaic array, where field confinement is achieved dynamically, leveraging the adjustable conductivity of the diodes under forward bias to turn the designated pixels into transient returns. We validated the computational modeling of the field confinement in such an optically-controlled circuit by in-vitro and in-vivo measurements. Most importantly, using this strategy, we demonstrated that the grating acuity with 40 µm pixels matches the pixel pitch, while with 20 µm pixels, it reaches the 28 µm limit of the natural visual resolution in rats. This method enables customized field shaping based on individual retinal thickness and distance from the implant, paving the way to higher acuity of prosthetic vision in atrophic macular degeneration.

Retinal degenerative diseases, such as age-related macular degeneration (AMD) and retinitis pigmentosa, are a leading cause of untreatable visual impairment and legal blindness[1,2]. Despite the irreversible loss of photoreceptors, the inner retinal neurons survive to a large extent[3–5], albeit with some remodeling[6,7]. Electrical stimulation of the second-order retinal neurons, mainly the bipolar cells, elicits visual percepts[8,9], hence enabling electronic restoration of sight (Fig. 1). AMD patients with a photovoltaic subretinal implant PRIMA (Pixium Vision, Paris, France) having bipolar pixels of 100 µm in width (corresponding to a Snellen acuity limit of 20/420) demonstrated a prosthetic letter acuity closely matching the pixel size: 1.17 ± 0.13 pixels, corresponding to the Snellen range of 20/438–20/565[8,10]. Even though this is an exciting

proof of concept, for a wide adoption of this approach by AMD patients, prosthetic acuity should significantly exceed their remaining peripheral vision, which is typically no worse than 20/400. The sampling limit for an acuity of 20/200 (US legal blindness threshold) corresponds to 50 µm pixels, while 20/100 requires 25 µm pixels.

As with natural vision, prosthetic visual acuity is fundamentally limited not only by the spatial resolution of the stimulation patterns (i.e. pixel size and the electric field spread in the retina), but also by their contrast, which is affected by crosstalk between the neighboring electrodes[11]. The lateral spread of an electric field can be confined by local return electrodes in each pixel, as in the PRIMA implant. However, scaling down such pixels is difficult because the penetration depth of

[1]Department of Physics, Stanford University, Stanford, CA, USA. [2]Department of Electrical Engineering, Stanford University, Stanford, CA, USA. [3]Hansen Experimental Physics Laboratory, Stanford University, Stanford, CA, USA. [4]Department of Ophthalmology, Stanford University, Stanford, CA, USA. [5]Department of Material Science, Stanford University, Stanford, CA, USA. [6]Department of Physics, Institute of Photonics, University of Strathclyde, Glasgow, Scotland, UK. [7]These authors contributed equally: Bing-Yi Wang, Zhijie Charles Chen. ✉e-mail: bingyiw@stanford.edu; zcchen@stanford.edu

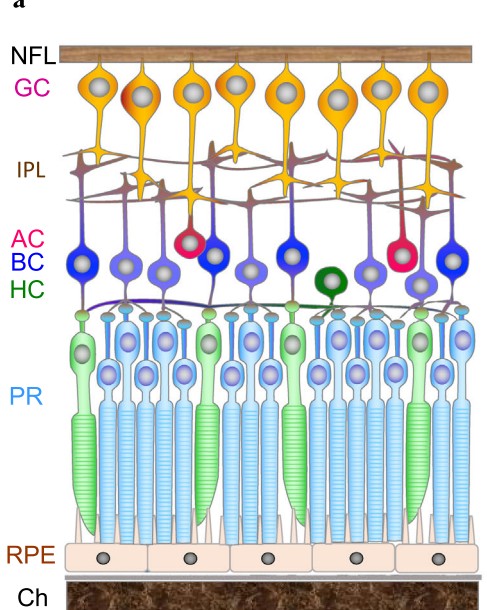
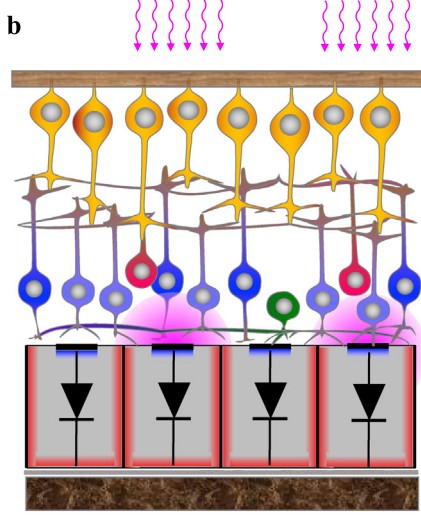

**Fig. 1 | Diagrams of the healthy and degenerate retinas. a** Diagram of the cellular layers in healthy retina, including the Choroid (Ch), Retinal Pigmented Epithelium (RPE), Photoreceptors (PR), Horizontal cells (HC), Bipolar cells (BC), Amacrine cells (AC), Inner Plexiform Layer (IPL), and Ganglion cells (GC), whose axons comprise the Nerve Fiber layer (NFL). **b** Degenerate retina with "electronic photoreceptors": a subretinal prosthesis composed of 20 μm wide and 30 μm deep photovoltaic pixels, which convert incoming light into electric current flowing through the tissue and polarize the nearby secondary neurons.

the electric field in tissue is constrained by the distance between the active and return electrodes, which is about half of the pixel radius (Fig. 2a). As a result, the retinal stimulation threshold in such a geometry rapidly increases with decreasing pixel size, approaching the charge injection limit of even one of the best electrode materials: sputtered Iridium oxide film (SIROF). Stimulation becomes insufficient for cortical measurements of acuity when pixel size falls below 55 μm in rats[12], and perceptual measurements of acuity below 75 μm in humans[13].

One approach to overcoming this problem is based on elevating the return electrode to the top of the inner nuclear layer using a 3-D honeycomb-shaped array, thereby orienting the electric field vertically within the wells (Fig. 2b)[12]. This arrangement decouples the field penetration depth from the pixel width and greatly reduces the stimulation threshold because the vertical field matches the orientation of the bipolar cells in the retina[12]. Although initial animal studies have shown promising results with retinal migration into the subretinal wells, functional activity of the migrated neurons into 3-D arrays has still to be confirmed. In addition, the fabrication process of the honeycomb structures with a local return electrode needs further development.

Here, we present an alternative approach to high-resolution prosthetic vision with planar monopolar subretinal implants[14]. Due to nearly vertical orientation of electric field in a full-field activation of such implants, the stimulation threshold is about 0.06 mW/mm² with 10 ms pulses−30 times lower than that with bipolar pixels of 40 μm in size (1.8 mW/mm²)[15]. Confinement of the electric field and high contrast with such monopolar arrays are enabled by the spatiotemporal modulation of the pixels, utilizing the exponentially increasing conduction of the photodiodes under forward bias, which drives the current in the direction opposite to the photocurrent.

In our photovoltaic approach to retinal prosthetics, images captured by a camera are projected onto the subretinal photovoltaic arrays from the augmented-reality glasses using pulsed near-infrared (880 nm) light[8,10]. Photodiodes in each pixel convert the incident photons into pulsed electric current flowing through the tissue between the active and return electrodes on the array. Bipolar cells polarized in this electric field, transmit the visual information via the retinal neural network to the brain, which allows preserving many features of the retinal signal processing, including the antagonistic center-surround, flicker fusion at high frequencies, nonlinear summation of the RGC subunits, and others[16,17]. This strategy is very different from the direct stimulation of the ganglion cells by epiretinal electrodes, which requires proper encoding of the spiking patterns for each cell type to reproduce the retinal signal processing[18], many aspects of which is not yet well-understood.

Accumulation of charge at the electrode-electrolyte interface and summation of the electric potentials from the neighboring pixels in electrolyte can elevate the voltage on the active electrodes. Since the active electrode is connected to the p+ contact of the diode and the return electrode is connected to the n+ contact in our pixels, the diode becomes forward biased when positive charge accumulates on interface of the active electrode with electrolyte. Therefore, pre-charging some of the active electrodes using proper illumination protocol can make them sufficiently conductive to serve as effective return electrodes for the next image frame, when these pixels are no longer illuminated, while others are activated. The distance between the active electrode and the neighboring return electrode affects the penetration depth of the electric field into the tissue (Fig. 2c). Such pre-conditioning of the pixels to become transient returns in the next image frame enables flexible control of the lateral and axial confinement of electric field in tissue by spatiotemporal modulation of the images projected onto the photovoltaic array. This optical current steering strategy potentially allows optimization of the stimulation depth and lateral selectivity for every patient, depending on the retinal thickness and its proximity to the implant[19].

To model this mode of operation, we first characterized the spatial coupling of electric potential among the pixels using a static finite-element method, and then computed the circuit dynamics of the photovoltaic array, taking into account the inter-pixel coupling, from which the dynamics of the potential distribution in the electrolyte is derived. The computational results are then validated by comparison to in-vitro measurements of the electric potential in electrolyte, as well as to electrograms recorded on the corneas of

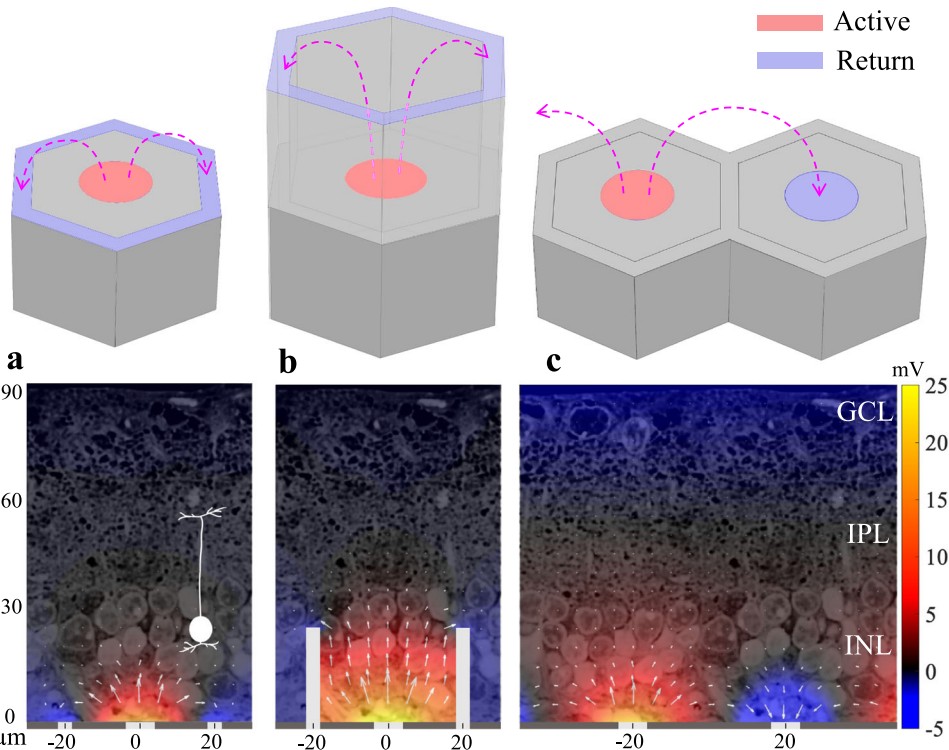

**Fig. 2 | Electric field with various implant geometries overlaid on retinal histology.** Electric field above the photovoltaic pixels (40 μm in width) of various geometries, with a potential relative to the middle of IPL (57 μm). **a** A bipolar pixel containing a central active and a circumferential return electrode. **b** A honeycomb-shaped pixel with elevated return electrode[12]. **c** Two monopolar pixels without local returns, where the active electrode of the left pixel acts as an anode and the one on the right—as a cathode, collecting 75% of the anodal current. The remaining current is collected by a global return around the edge of the device. The bottom panels show the corresponding simulated electric fields overlaid on a histology image of a degenerate rat retina (current = 68 nA). Diagram of a bipolar cell with its axonal terminals in the middle of IPL in the left panel illustrates its position and size with respect to the field penetration depth. The arrows represent current magnitude on a log scale. Source data are provided as a source data file.

rats implanted with photovoltaic arrays. Most importantly, we demonstrate for the first time that prosthetic acuity in blind rats with subretinal implants composed of 40 μm pixels matches the pixel pitch, while with 20 μm pixels, it reaches the 28 μm limit of their natural visual resolution.

## Results

### Computational modeling of the electric field in tissue

The charge accumulated at the electrode-electrolyte interface of a photovoltaic pixel discharges by forward-biased conduction through the diode, driven collectively by the potential at the electrode-electrolyte capacitance due to charge accumulation and the effect of electric potentials from the neighboring pixels. When the forward current through the diode becomes comparable to the stimulation current of the neighboring pixels, such a discharging pixel serves as a transient return. Such a local return electrode limits the spread of electric fields from adjacent pixels, reducing the crosstalk and limiting the field penetration into the tissue (Fig. 2c). The cross-pixel resistance between two adjacent electrodes can be defined as $R_{1,2} = V_{1,2}/I_1$, where $I_1$ is the current injection of electrode 1, and $V_{1,2}$ is the potential rise at electrode 2 caused by $I_1$. The finite-element model (COMSOL) of the subretinal photovoltaic array with 40 μm pixels yields $R_{1,2}$ in the range of 15 to 30 kΩ, depending on the location of the electrodes in the array. Each pixel in the hexagonal mesh is surrounded by 6 neighbors, and the stimulation current is typically between 0.1 and 1 μA. Therefore, summation of these electric potentials in the electrolyte during simultaneous activation of multiple pixels, together with accumulation of charge at the capacitive electrode-electrolyte interface during previous pulses, may elevate the electrode potential close to the turn-on voltage of the Si diode (about 0.5 V). A more comprehensive description of the circuit dynamics is provided in the Supplementary Materials.

Figure 3 shows the effect of such transient returns on a grating pattern, where every other line of the 40 μm pixels is activated by illumination at 1 mW/mm². In this calculation, we assumed exact alignment of the light pattern to the pixels. Misalignment will cause some smearing and decrease in contrast, varying over time due to the eye movements. Since the activating function of the bipolar cell stimulation can be closely approximated by a trans-cellular voltage step (between the dendritic and axonal ends of a neuron)[20], we plot a potential difference between 10 μm above the implant (bottom of the BC dendrites) to 57 μm—the middle of the IPL, where the axonal terminals of the average BC are located[12], as illustrated in Fig. 2a. When the electrode-electrolyte interfaces of the dark pixels are pre-charged to 0.5 V prior to the light pulse, they become sufficiently conductive to drain the current during the next pulse and hence the potentials in the electrolyte above the dark pixels drop below 0 V, thereby providing 100% contrast (Fig. 3b, d). On the contrary, if the electrochemical potential bias is far below the turn-on voltage of the diode (e.g. 0.2 V) the dark pixels are not conductive and hence do not transform into transient returns. Therefore, the contrast is reduced to below 30% as a result of a strong crosstalk from the neighboring pixels (Fig. 3b, c). Contrast decreases even further if the array is composed of smaller pixels, and it also declines with distance from the implant. Such low contrast precludes high-resolution prosthetic vision in general, and grating acuity measurements, in particular.

### Experimental validation of the model

We recorded the electric potential, generated by the photovoltaic array in Dulbecco's phosphate-buffered saline (DPBS) under 10 ms NIR

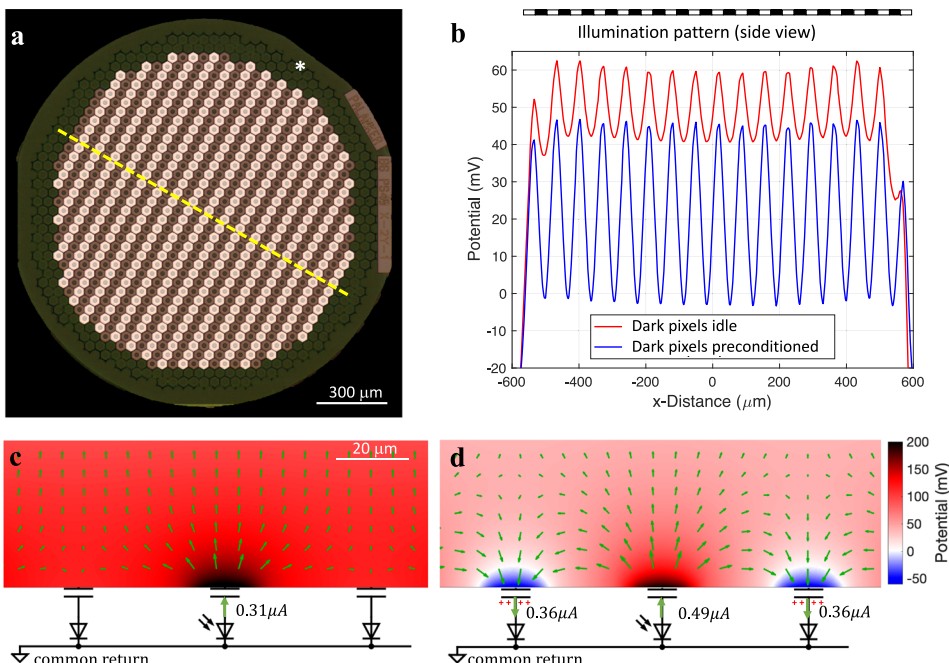

**Fig. 3 | Modeling the effect of preconditioning the dark pixels.** Electric field in electrolyte generated by the grating pattern with and without preconditioning of the dark pixels. **a** Photovoltaic array of 1.5 mm in diameter composed of monopolar pixels of 40 μm width, with a common return electrode (indicated by *) along the edge of the implant. **b** Electric potential in electrolyte 10 μm above the implant relative to the middle of IPL (57 μm), plotted along the dash line shown in **a**, when the dark pixels are non-conductive (red) and preconditioned (blue). **c**, **d** Magnified view of the electric potential and current in the medium for three adjacent pixels. Length of the arrows represents the 0.6 power of the magnitude of the current density for visualization. Scale bar is 20 μm. Source data are provided as a source data file.

(880 nm) pulses repeated at 40 Hz, using a pipette electrode positioned 20 μm above a pixel in a Petri dish. First, the device was driven to the steady-state under uniform full-field illumination at a peak irradiance of 8 mW/mm² (Fig. 4a). Then a field stop in the projection system was rapidly closed between the two laser pulses, restricting the illumination to an octagonal area of about 1 mm in width, as shown in Fig. 4b, c. The electric potential outside the field stop was recorded during the transition into the post-closure steady-state. Although the recorded pixel in the shadowed area does not inject current after the closure, positive pulses are still detectable in the post-closure steady-state due to crosstalk from the illuminated pixels. However, during the transition period between the field stop closure and the post-closure steady-state, instead of a positive pulse, a characteristic fast negative dip was observed at the onset of the first light pulse after the closure (Fig. 4d, *t* = 0 ms), in addition to the larger and slower negative swing caused by the current flowing to the large transient return. These transients demonstrate that current is sinking through the discharging dark pixels, boosted by the spatial coupling of electric potentials in electrolyte, as predicted by the model.

With alternating gratings, the field confinement effect of the transient returns should be stronger at higher alternating frequency since the shorter discharge time between the pulses is associated with the increased forward bias, resulting in stronger conduction of the transient returns and hence weaker far-field potential. To validate such prediction of the model, we calculated the electrical signal generated by the implant at the cornea for various alternating frequencies, ranging from 1 to 125 Hz. For this purpose, we created a computational model of the rat eye[21] and a head, with a photovoltaic array in the subretinal space (Fig. 5a), as described in detail in Methods and in Supplementary Materials. The modeling results were compared with the in-vivo measurements of the corneal potential. With a full-field illumination at 1.2 mW/mm² of 4 ms in duration repeated at 1 Hz, the amplitude of the corneal potential was in the range of 2 to 3.5 mV, varying between animals and with the position of the electrode on the cornea. As shown in Fig. 5b, modeling closely matches the shape of the measured corneal signal. Due to the variation of the corneal signals across animals, we normalized the frequency dependence plot by the amplitude at 1 Hz before averaging. As shown in Fig. 5c, the frequency dependence of the corneal signal also closely matches the model. Amplitude of the visually evoked potentials (VEPs) decreases with increasing frequency much faster than the corneal signals, dropping to the noise level near 60 Hz (flicker fusion). For this reason, we performed the acuity measurements at a pulse repetition rate of 64 Hz, as described in the next section.

### Grating acuity in natural and prosthetic vision

Measurements of the VEPs in response to alternating gratings is a common method of assessing vision in animals[22] or infants[23]. We recorded VEPs via transcranial electrodes above the visual cortices in Royal College of Surgeons (RCS) rats having subretinal photovoltaic implants, and in Long Evans (LE) rats, as a normally-sighted control. We experimentally confirmed that these RCS rats did not exhibit VEP response before implantation, nor when the light was projected outside the implants. Corneal signal generated by the implant was also recorded for characterization of the implant's current injection. For prosthetic vision, grating images were delivered with NIR light (880 nm) at 1.2 mW/mm² peak irradiance at the retina using 4 ms pulses at 64 Hz repetition rate, while the pattern reversal occurred every 500 ms (Fig. 6). We evaluated the prosthetic visual acuity with implants having 40 μm and 20 μm pixels. For the natural visual acuity measurements, images of the black-and-white gratings presented on a screen under continuous illumination also alternated every 500 ms.

The prosthetic VEP waveforms contained both a 2-Hz signature induced by the grating reversal, and a 64-Hz component of the carrier frequency, including the stimulus artifact. We used the corneal signal as a template to remove the artifact and filtered out the high frequency components. Amplitude of the grating reversal response was measured as the peak-to-peak voltages in the VEP waveform between 0 and

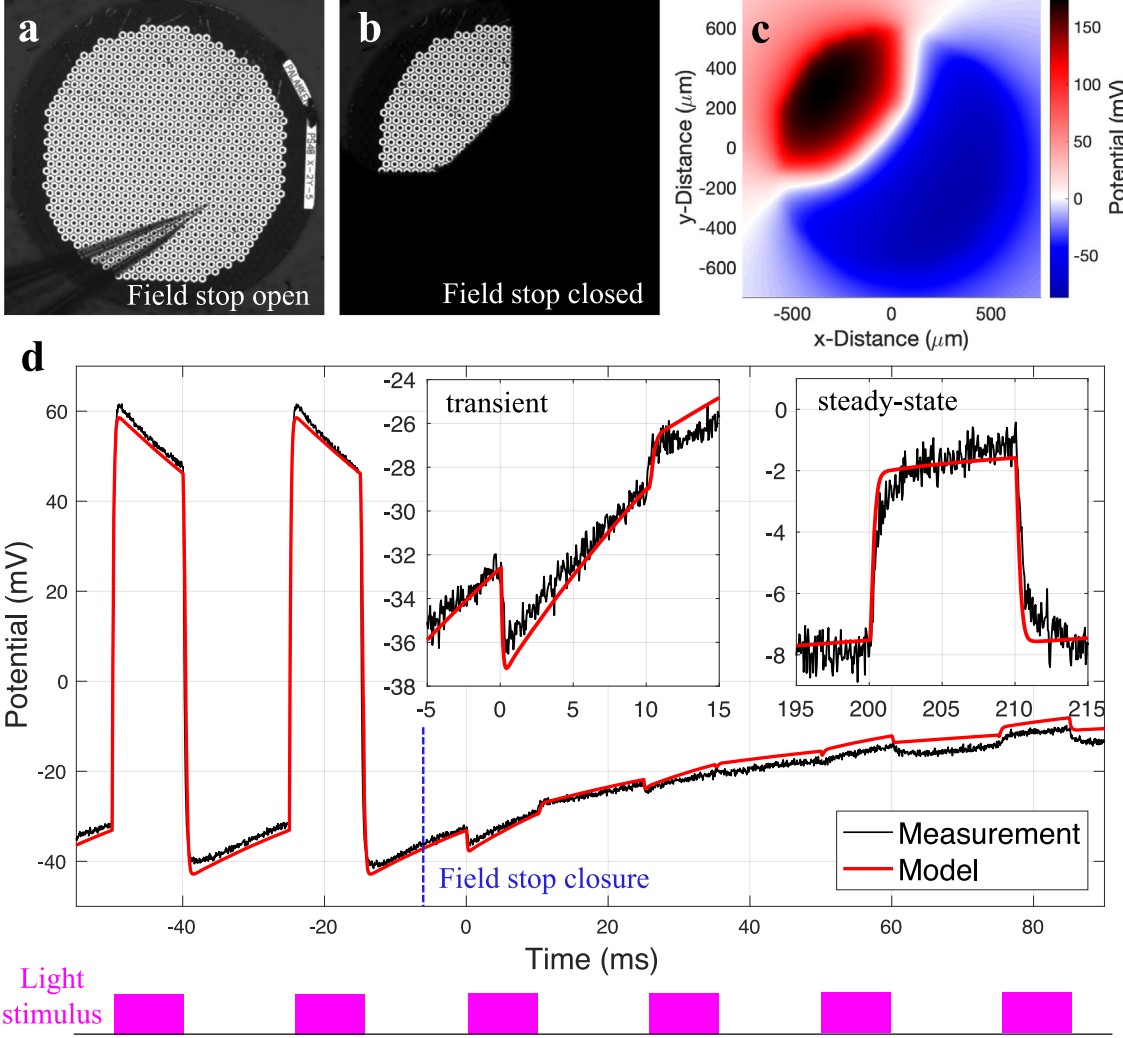

**Fig. 4 | In-vitro validation of the model.** Near-field electric potential in electrolyte 20 μm above the photovoltaic array. **a** View of the array and a pipette electrode under full-field illumination. **b** The partially illuminated array after closure of the field stop. The pipette is at the same location, in the shaded zone. **c** Map of the electric potential at 0.6 ms. **d** Electric potential induced by 10 ms pulses of NIR light before (<0 ms) and after (>0 ms) the decrease of the illuminated area. Source data are provided as a source data file.

100 ms. The natural grating acuity is typically defined in angular frequency units, i.e. cycles per degree (cpd), and is determined in electrophysiological measurements by the intersection of the logarithmic fitting line with the noise level in a log-linear plot[24,25]. We followed this convention (Fig. 7), but to relate the resolution to the pixel size, we plotted the VEP amplitude vs. the inverse bar width (half a cycle). In this conversion, we utilized the fact that one degree of the visual angle corresponds to $64.3 \pm 2.9$ μm on the retina, as described in Methods. Corresponding plots in the linear units of the bar widths are shown in Supplementary Fig. S6.

For prosthetic vision with 40 μm implants, the visual acuity was found to be $34.3 \pm 4.0$ μm, matching the pixel pitch in a hexagonal array $p = 40$ μm·cos (30°) = 34.6 μm. However, with 20 μm pixels the grating acuity did not match the row pitch of 17.3 μm. Instead, it was $27.5 \pm 4.0$ μm, matching the natural grating acuity of $27.9 \pm 3.0$ μm, measured in LE rats, as shown in Fig. 7. The average natural visual acuity of 27.9 μm corresponds to 1.15 cpd, closely matching the literature[24,26–28]. For all three plots, the uncertainty range of the visual acuity corresponds to statistical significance with $p < 0.05$. If the average noise level across all animals ($8.89 \pm 2.52$ μV) would be used instead of the noise ranges averaged within each group (Fig. 7), the acuity would be 38.2, 26.1 and 27.2 μm for pixels of 40 and 20 μm and

for natural vision, respectively, i.e. within the error bars defined by the noise level in each group. In Fig. 7 and S6, the shown acuity values are rounded to the nearest integer.

## Discussion

All previous studies with subretinal photovoltaic implants showed that the prosthetic visual acuity matches the pixel pitch (100 μm in humans[8], 75 and 55 μm in rats[15,16]), just as the case with our 40 μm pixels in this study. Here, however, we demonstrated that prosthetic visual acuity with 20 μm pixels is not limited by the pixel pitch, but rather matches the 28 μm natural visual acuity limit in rats, indicating that with such small pixels, resolution appears to be limited by the retinal network integration rather than by the pixel size, although other unknown limiting factors cannot be excluded. In humans, where the 20/20 vision corresponds to 5 μm stripes on the retina, the 20 μm pixels might enable restoration of central vision in AMD patients with acuity as high as 20/80, which would be of major clinical significance. Natural resolution becomes worse than 20 μm at eccentricity exceeding about 7 degrees[29], or 2 mm from the fovea. Therefore, the centrally placed implant smaller than 4 mm in diameter with 20 μm pixels is not expected to exceed the natural resolution in human retina. It is important to keep in mind that retinal rewiring in AMD patients[6]

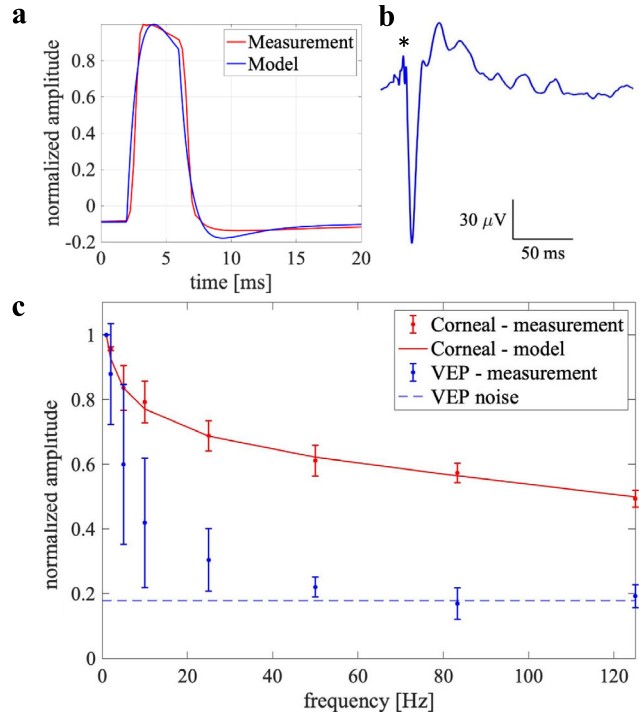

**Fig. 5 | In-vivo validation of the model.** Far-field electric potential in the eye: measurements and modeling. **a** Measured and modeled corneal potential generated by a 4 ms pulse at 1.2 mW/mm² full-field illumination. **b** Visually evoked potential in response to 4 ms pulses applied at 5 Hz repetition rate. Asterisk indicates the 4ms-long positive stimulus artifact. **c** Peak-to-peak amplitudes of the corneal signal and VEP induced by the 4 ms pulses at repetition rates varying from 1 to 125 Hz under full-field illumination, normalized to 1 Hz. Modeling result for the corneal signal is shown by a line. For both corneal and VEP measurements, $n = 3$ independent experiments. The error bars represent the standard deviations with centers at mean values. Source data are provided as a source data file.

may further limit the fidelity of prosthetic vision. In addition, sub-retinal debris in AMD patients, separating the INL from the implant by about 30–40 µm[8] may impose additional constrains on attainable resolution, compared to rodents[30].

An electric field around the pixels could be optimally confined by adjusting the distance between the active and return electrodes. Such optical configurability of the electric field enables optimization of the field penetration depth for individual patients, tailored to their particular retinal thickness and the implant's proximity to bipolar cells[19]. At the same time, properly restricting the field penetration depth may avoid the direct activation of the tertiary retinal neurons: the amacrine and ganglion cells. Ideally, this could be accomplished if each pixel would have an independent control of its conductivity at any moment in time. This could be done, for example, by including a photosensitive transistor connected in parallel with a primary photodiode in each pixel, which would respond to a wavelength different from the primary NIR light. However, this would greatly complicate the implant design and fabrication. Here we describe a much more practical approach to such field confinement.

The monopolar photovoltaic array having a common return electrode allows the use of some of the active electrodes as anodes and some as cathodes, depending on their pre-charging history and the light intensity on each pixel. Due to charge accumulation during previous exposures, the electrode potential stays close to the diode turn-on voltage, so its forward current balances the photocurrent corresponding to the average light intensity. The charge accumulation and the spatial coupling of electric potential between the electrodes collectively determines the cathodal current amplitude of a transient return.

Rapid computation of the current steering sequence for generating a target electric field in the retina based on the images captured by the camera can be performed using an optimization-based approach[19]. For a typical image refresh rate of 30 Hz, each frame lasts about 33 ms. Photovoltaic stimulation pulses typically range from 0.8 to 10 ms[8], leaving at least 23 ms for the pre-charging phase preceding the stimulation phase in each frame. Upon image acquisition, the sufficiently dark pixels will be designated to become the local returns and will be exposed to light below the stimulation threshold to accumulate additional charge prior to activation of the bright pixels in the image. Then, during the stimulation phase, the brighter pixels are illuminated with intensity and duration corresponding to the desired charge injection, while the pre-charged dark pixels sink the current injected from the nearby active pixels. This pre-conditioning strategy has the potential to achieve the field confinement and hence a high spatial resolution, with a delay not exceeding one frame relative to the image acquisition. To accelerate the image processing for determining which pixels need to be pre-conditioned, predictive tracking algorithms, such as the Kalman filter or exponential smoothing, could be applied. This approach may also allow stretching the pre-charging phase over several frames.

Another strategy for reducing the crosstalk could be a sequential stimulation spread over the whole 33 ms of the frame duration. With the average pulse duration of about 4 ms, only about 4/33 = 12% of the pixels can be activated simultaneously, and even fewer than that, if the images are sparse. In such a distributed activation, pixels become the transient returns after their light is turned off, while other pixels are turned on. However, with a stimulation spread over the frame duration, eye movements may shift the projected image and introduce some uncertainty regarding the relative location of the active and return electrodes. During the slow drift (<0.25 deg/s), the 2.5 µm/frame image shift is negligible compared to the 20 µm pixel size, but it becomes noticeable during microsaccades, which occur 1–2 times a second[31]. On average, the eye movement velocity is about 1 deg/s[32], corresponding to about 10 µm/frame at 30 Hz frame rate, which imposes a lower limit on the pixel size for predictable activation, unless eye tracking is employed or frame rate is increased to reduce this uncertainty.

In conclusion, we have demonstrated prosthetic vision with a resolution matching the natural acuity in rats, using a novel design of a photovoltaic array, where electric field confinement is achieved dynamically, by turning the designated pixels into transient returns. This method enables customized field shaping for each patient and paves the way to prosthetic vision with acuity exceeding 20/100.

## Methods

All experimental procedures were conducted in accordance with the Statement for the Use of Animals in Ophthalmic and Vision research of the Association for Research in Vision and Ophthalmology (ARVO) and approved by the Stanford Administrative Panel on Laboratory Animal Care (APLAC protocol 13765 and 33394). Royal College of Surgeons (RCS) rats were used as an animal model of inherited retinal degeneration, and Long Evans (LE) rats were used as a control group with healthy natural vision. Animals are maintained at the Stanford Animal Facility under 12 h light/12 h dark cycles with food and water ad libitum.

### Modeling and in-vitro validation of the implant performance

We modeled the electric field in the retina generated by a subretinal 30 µm-thick array 1.5 mm in diameter with 40-µm photovoltaic pixels, described in detail in[14]. In this array, each pixel has a SIROF-coated active electrode of 18 µm in diameter, exposed to electrolyte, and a photosensitive area insulated by SiO₂ and SiC. The return electrodes are connected to a SIROF-coated common return deposited at the periphery of the implant, as shown in Fig. 3a. The electric field at any given time is a linear combination of a set of elementary fields, each of

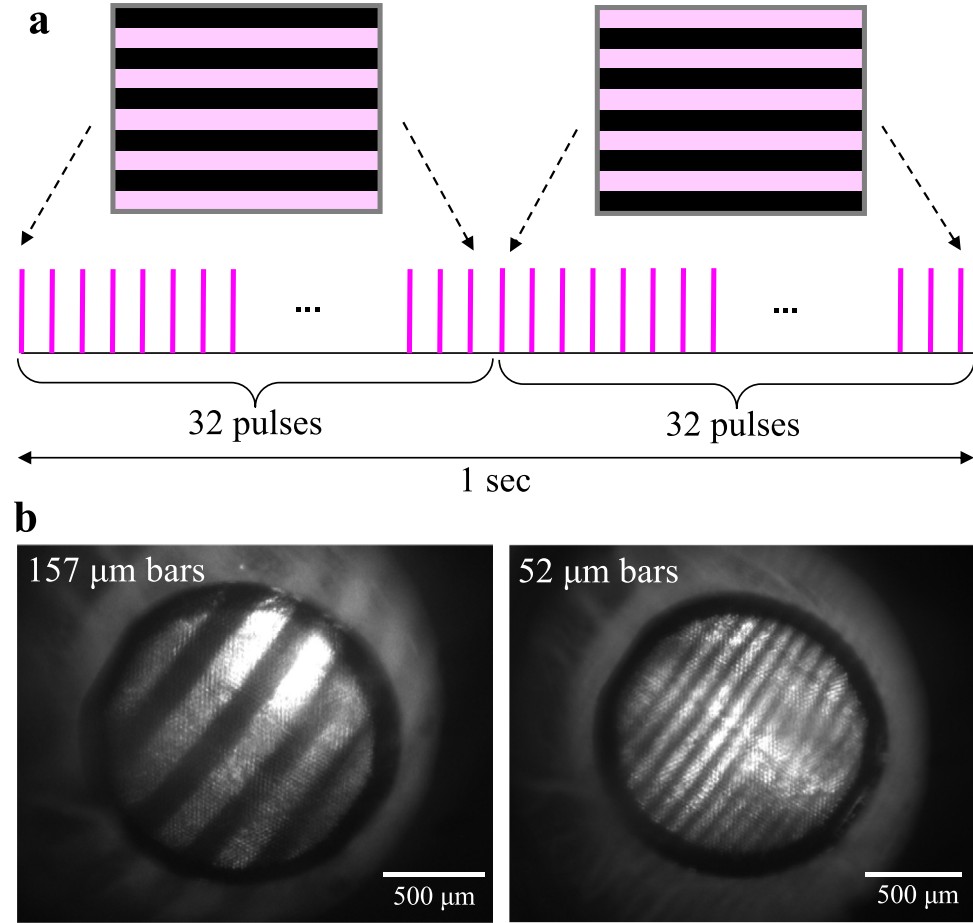

**Fig. 6 | Measurement scheme of the grating acuity. a** Projected grating patterns alternate at 2 Hz, pulsed at a carrier frequency of 64 Hz, with a pulse duration of 4 ms. **b** Images of the projected gratings with bar widths of 157 and 52 µm, respectively, on a 20 µm implant in a rat eye.

which corresponds to activation of one pixel individually, with coefficients equal to the currents at the respective active electrodes[19]. We generated the elementary electric fields of all pixels by numerically solving the Poisson equation of volume conduction, using the finite element method (FEM) in COMSOL Multiphysics 5.6 (COMSOL, Inc., Sweden), the model file of which is accessible online[33]. The boundary condition on the capacitive electrode-electrolyte interface of a SIROF electrode changes from equipotential (EP) to a uniform current density (UCD)[34], with a time constant of t ≈ RC, which is 0.24 ms for the active electrodes and 40 ms for the common return electrode. Since the pulse width in our measurements ranges from 4 ms to 10 ms, we assumed the UCD boundary condition for the active electrodes and EP - for the return.

To model the spatial coupling of the electric potential between pixels, we generalized the access resistance of one electrode to the cross-resistance matrix **R** of all active electrodes, where the entry in row $i$ and column $j$ corresponds to the potential elevation at an active electrode $i$, caused by a unitary current injection from an active electrode $j$[19]. The diagonal entries of **R** represent the access resistance of each active electrode. To compute the interrelated circuit dynamics of all pixels, we then generalized the equation of the nonlinear circuit dynamics of a photovoltaic pixel[35] to the multi-dimensional form, as shown in Supplementary Materials. The multi-dimensional nonlinear differential equation was solved with our customized adaptive-step gradient descent method in MATLAB R2017a (MathWorks, MA), the source code of which is accessible online[36]. To compute the electric field in the retina, the elementary electric fields were linearly combined using coefficients obtained from the multi-dimensional circuit dynamics.

For validation of the spatiotemporal model, we placed the device in a Petri dish filled with Dulbecco's phosphate-buffered saline (DPBS) diluted to 10% by volume to mimic the retinal resistivity. The 10-ms pulses of 880-nm laser light were projected onto the device surface at a repetition rate of 40 Hz at intensity of 8 mW/mm². Micropipette with a 5 µm tip diameter was used to measure the electric potential 20 µm above the device versus a Ag|AgCl reference wire a few centimeters from the device. To study the current dynamics under changing images, a field stop in the laser projection system was swiftly closed between two laser pulses, reducing the illuminated area from full-field to an octagon covering about a quarter of the pixels (Fig. 4a). Electric potential before and after the closure was modeled and compared to the measurement. Due to the drifting electrochemical potential of the reference wire, the modeling result were shifted down by 2.9 mV to match the measured average potential in the steady state.

**Modeling the corneal signals and the in-vivo validation**
To model the electric potential generated by a subretinal implant on the cornea, we constructed an anatomically realistic model of the rat eye as shown in Figure S4b in Supplementary Materials, similar to the model by Selner et al.[21] but with adaptation for the thickness of a degenerate retina and a photovoltaic implant in the subretinal space. We also incorporated a realistic geometry of the rat head to the model and computed the potential difference between the corneal and the nasal electrodes, as shown in Figure S4a in Supplementary Materials (COMSOL model file is accessible online[33]). Meshing and adjusting the complex geometry of the rat head was difficult and computationally slow, and since the initial result demonstrated that electric potential is confined to a few millimeters around the eye, we approximated the

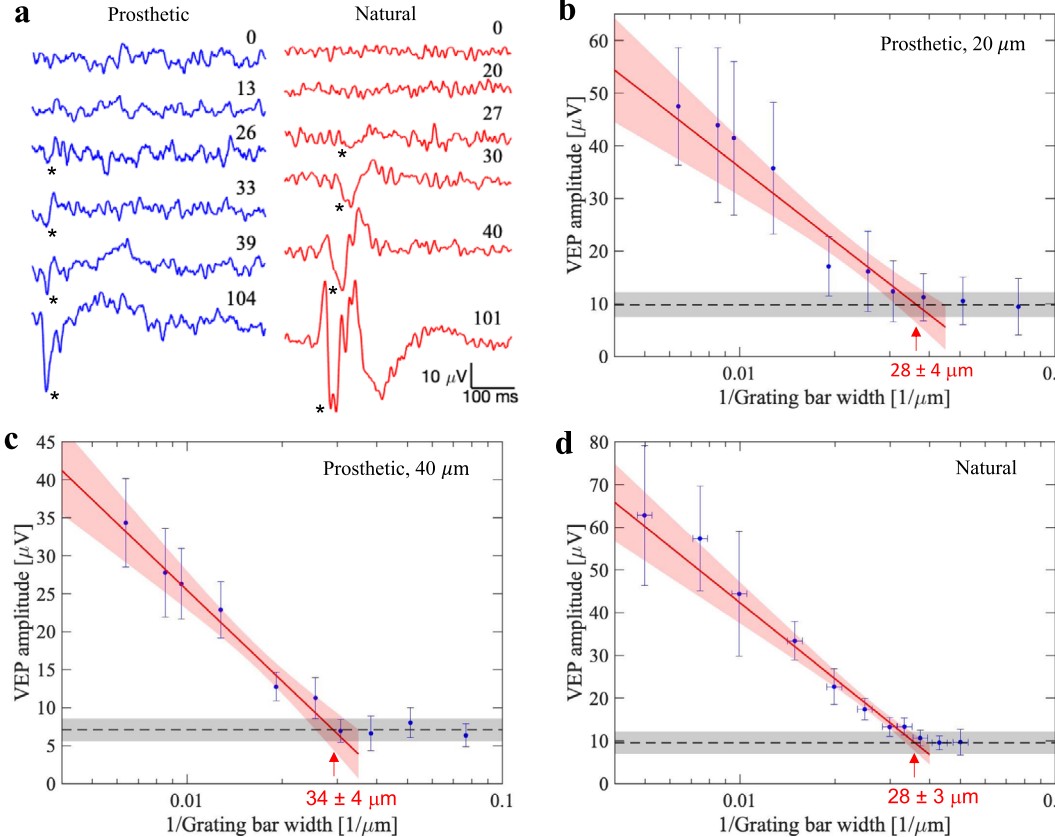

**Fig. 7 | Prosthetic and natural visual acuity in rats. a** Example VEP waveforms in response to alternating gratings in RCS rats (blue) with implants and in LE rats (red). Grating bar width in μm is shown next to each waveform. Asterisk points at the N1 peak. **b**–**d** VEP amplitude as a function of the grating density (in units of the inverse bar width, 1/μm) for prosthetic vision with 20 μm pixels **b**, 40 μm pixels **c**, and for natural vision **d**. The error bars represent the standard deviations centered at mean values, which are derived from $n = 5$ **b**, $n = 4$ **c**, and $n = 6$ **d** biologically independent animals, respectively. The vertical error bars indicate the standard deviations across animals, while the horizontal error bars in **d** represent the uncertainty in conversion from angular units of resolution (cpd) to linear bar width on the retina (μm) (see Methods for further details). The black dashed lines represent the mean noise level, while the gray bands indicate the noise spread among animals. Red line is a logarithmic fit, which defines acuity as a crossing point with the noise level, pointed by the arrow. Red band around the fit line represents the 95% confidence interval. Source data are provided as a source data file.

rest of the head geometry with a cylinder, while keeping all the ocular and orbital tissues the same as in the anatomically accurate model. This approximation resulted in a similar electric field, but with a more reliable meshing and adjustable geometry. Electric potential generated by a single activated pixel in such a simplified layout is shown in Fig. 5a, overlaid with the diagram of the ocular anatomy. A more detailed diagram of the anatomical structures and the list of the tissue properties with the corresponding references are shown in the Supplementary Materials.

**Surgical procedures**

The photovoltaic devices were implanted in the subretinal space, as previously described[37], typically at 6 months of age, after a complete loss of the outer nuclear layer, as evidenced by optical coherence tomography (OCT; HRA2-Spectralis; Heidelberg Engineering, Heidelberg, Germany). The implants were placed in the temporal-dorsal region, approximately 1 mm away from the optic nerve. A total of 9 animals were implanted with 1.5 mm diameter arrays containing pixels of 40 μm ($n = 4$) and 20 μm ($n = 5$). Animals were anesthetized with a mixture of ketamine (75 mg/kg) and xylazine (5 mg/kg) injected intraperitoneally.

To visualize the retina and the implant, animals were monitored over time using OCT (see Supplementary Fig. S5 for a typical set of OCT images before and after implantation). For measurements of the visually evoked potentials (VEP), each animal was implanted with three transcranial screw electrodes: one electrode over each hemisphere of

the visual cortex (4 mm lateral from midline, 6 mm caudal to bregma), and a reference electrode (2 mm right of midline and 2 mm anterior to bregma).

**Measurements of the prosthetic visual acuity**

Acuity measurements were first performed within an hour post-implantation, and then repeated over several months of the follow-up period. Following anesthesia and pupil dilation, the cornea was covered with a viscoelastic gel and a cover slip to cancel its optical power and ensure good retinal visibility. The retinal implant was illuminated with a customized projection system, consisting of an 880 nm laser (MF-880 nm-400 μm, DILAS, Tucson, AZ), customized optics, and a digital micromirror display (DMD; DLP Light Commander; LOGIC PD, Carlsbad, CA) for pattern projection. The entire optical system was integrated with a slit lamp (Zeiss SL-120; Carl Zeiss, Thornwood, NY) for real-time observation of the illuminated retina via a CCD camera (acA1300-60gmNIR; Basler, Ahrensburg, Germany).

Visual acuity was assessed by applying alternating gratings, where the NIR illumination was pulsed at 64 Hz and the grating patterns were switched every 32 pulses (with a period of 1 s). The grating bar width ranged from 13 μm to 157 μm on the retina, while the light intensity was set to 1.2 mW/mm². The light intensity at the cornea was measured before and after each session and scaled by the ocular magnification squared, where magnification is defined as the ratio between the size of a projected square on the retina and in air. VEPs were recorded using the Espion E3 system (Diagnosys LLC, Lowell, MA) at a sampling rate of

2 kHz and averaged over 250 trials. Corneal signal reflecting the device performance was simultaneously measured using ERG electrodes, relative to the reference electrode in the nose. The corneal signal was also used as a template to remove the stimulus artifact in VEP waveforms. High frequency components in VEP traces were further filtered using a spectrum reconstruction method.

To determine the noise floor, we projected static gratings of 120 μm and 30 μm in widths at the same light intensity and pulsing frequency. After removing the stimulus artifacts at carrier frequency, we measured the peak-to-peak amplitude during the first 100 ms after the gratings' onset, identical to the way we define amplitude with alternating gratings. These measurements, averaged across animals, yielded the average noise level plotted as black dashed lines in Fig. 7. The noise distribution is shown by a gray band, whose upper and lower bounds are set at one standard deviation away from the noise mean value.

Acuity was defined as an intersection of the logarithmic fitting line with the noise level–the center value of acuity is the intersection with noise mean and the error bar is derived from the variances of noise distribution and uncertainties associated with the curve fit. The data points used for the logarithmic fit were the ones whose amplitude exceeded the noise level, as verified by the unpaired t test relative to the noise. To assess the statistical significance of the acuity range defined by the fit, we extracted the uncertainties in the fitted parameters from the covariance matrix and found the acuity range corresponding to the 95% confidence interval (corresponding to $p = 0.05$) associated with the uncertainty of the fit. These VEP data were analyzed using custom code developed in MATLAB and Python.

Electrophysiological measurements included multiple sessions, demonstrating safety of the retinal stimulation even with the smallest pixels. Photoresponsivity of our photodiodes is 0.51 A/W[14], and the photosensitive area per pixel is about twice larger than the active electrode. Therefore, the maximum charge injection density on electrodes at 1.2 mW/mm$^2$ irradiance with 10 ms pulse is about 1.2 mC/cm$^2$, which with the SIROF capacitance[14] of 6 mF/cm$^2$ corresponds to the voltage step of about 0.2 V–well within the water window of −0.6 to +0.8 V versus Ag|AgCl[38]. In fact, with a single Si diode per pixel, the photovoltage is limited to 0.6 V, and distributed between the active, the return electrodes and the ohmic voltage drop in the medium, cannot exceed the water window in principle. Clinical safety of the photovoltaic subretinal implants PRIMA has been demonstrated under 3.5 mW/mm$^2$ irradiance with 10 ms pulses at 30 Hz rep. rate[8,10]. These bipolar pixels are composed of two diodes connected in series, generating the maximum voltage of about 1.2 V. Since the new monopolar design includes only one diode per pixel, the maximum voltage and the maximum charge density is twice lower than with the 2-diode pixels, and hence can only be safer.

### Measurements of the natural visual acuity

There is a range of values for rats' natural acuity in the literature. Since the VEP amplitude and the noise level depending on the setup, we wanted to make sure our comparison between the prosthetic and natural vision will be performed with the same electronics, anesthesia, data analysis, etc. Since our NIR projector exhibited strong chromatic aberrations at visible wavelengths, resulting in a significant loss of contrast with fine patterns, we measured the natural acuity in LE rats ($n = 6$) using a conventional white screen in angular units (cycle per degree), and then related it to the linear resolution (μm) on the retina.

Following anesthesia, the cornea was covered with a plano contact lens and animals were placed 23 cm away from a 40 cm-wide white screen, providing luminance of 51 cd/m$^2$ on the cornea. Black-and-white rectangular-wave grating patterns of 50% duty cycle alternated at 2 Hz, providing 500 ms exposure duration for each pattern. Stripe widths varied from 0.21 to 16.7 mm on the screen, corresponding to

9.6 to 0.12 cycles per degree (cpd) in the eye. VEP measurements were averaged over 100–150 cycles.

To relate the angular units of acuity (cpd) to the stripe width on the retina (μm), we performed spot retinal photocoagulation (PASCAL laser, Topcon, San Jose, CA). Animals were anesthetized and mounted on a gonioscopic stage, and retinal laser burns were produced for 3 angular positions separated by 4 degrees each. The 577 nm wavelength laser with a spot diameter of 100 μm was centered on the pupil for each exposure of 70 ms in duration at 70 mW of power (or two sequential pulses of 100 ms at 50 mW). Retinal burns were then imaged together with the NIR grating pattern projected into the eye in our slit-lamp based VEP system used for the measurements of prosthetic acuity, as described above. Distance between the centers of the burns was measured using the NIR patterns as a scale bar. The resulting conversion factor was found to be $64.3 \pm 2.9$ μm per degree of the visual angle, very similar to the $65.3 \pm 6.7$ μm/deg based on optical modeling of the rat eye[39]. The uncertainty range in this conversion is represented by the horizontal error bars in Fig. 7d. Logarithmic line was fit to the falling trend of the plot, excluding the edge data point corresponding to the maximum value. Acuity was defined in the same way as described above for prosthetic vision.

### Statistics and reproducibility

To determine the confidence interval of the acuity limits in rats, we first fit the VEP amplitudes to a linear function of grating width on a logarithmic scale (Fig. 7), with a curve_fit() function from optimization module of Scipy (version 1.7.1) in Python 3.8.8, which reports the covariance matrix of the fit parameters. The variance of the VEP fit value was then calculated as a function of the grating width. At the intersection between the VEP fit line and the noise level, defined as the nominal acuity limit, we used the statistical delta method to find the standard deviation of the acuity limit from those of the fit line and the noise level. By the normal approximation, the 95% confidence interval of the acuity limit was then determined as 1.92 times its standard deviation on each side of the intersection. The detailed derivation is in Section 3 of the Supplementary Materials.

### Reporting summary

Further information on research design is available in the Nature Research Reporting Summary linked to this article.

## Data availability

Source data are provided with this paper. The in-vivo VEP data and in-vitro data that support the findings of this study are available from the corresponding author upon reasonable request. All other relevant data supporting the key findings of this study are available within the article and its Supplementary Information files or from the corresponding author upon reasonable request. Source data are provided with this paper.

## Code availability

The custom computational model and code are available on Zenodo, with access links https://doi.org/10.5281/zenodo.5081278 and https://doi.org/10.5281/zenodo.5081286, respectively.

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

## Acknowledgements

Studies were supported by the National Institutes of Health (Grants R01-EY-027786, B.W., Z.C., M.B., T.H., A.S., V.Z., E.H., L.G., T.K., D.P., and P30-EY-026877), the Department of Defense (Grant W81XWH-19-1-0738, B.W., Z.C., M.B., A.S., E.H., L.G., T.K., D.P.), AFOSR (Grant FA9550-19-1-0402, B.W., Z.C., M.B., T.H., A.S., V.Z., E.H., L.G., T.K., D.P.), Wu Tsai Institute of Neurosciences at Stanford (T.H., A.S.), and unrestricted grant from Research to Prevent Blindness. Photovoltaic arrays were fabricated at the Stanford Nano Shared Facilities (SNSF) and Stanford Nanofabrication Facility (SNF), which are supported by the National Science Foundation award ECCS1542152. K.M. was supported by a Royal Academy of Engineering Chair in Emerging Technology, UK. We would like to thank Dr. Tong Ling (currently at Nanyang Technological University, Singapore) for his help with the design and assembly of the optical system for prosthetic acuity measurements and Dr. Felix Abramovich (Tel Aviv University, Israel) for consulting on statistical analysis of the visual acuity.

## Author contributions

B.-Y. Wang conducted the electrophysiological measurements, Z.C. Chen performed the computational modeling and in-vitro characterization of the implant, M. Bhuckory performed the implantations and in-vivo imaging and helped with prosthetic acuity measurements, T. Huang, A. Shin, and L. Galambos worked on the fabrication of the implants under the guidance of T. Kamins and K. Mathieson. V. Zuckerman conducted natural acuity measurements, E. Ho built that system and helped with the software development for natural acuity measurements. Ethan Rosenfeld worked on the computational model of the eye

and head in COMSOL. D. Palanker guided the research and data analysis. All authors participated in writing the manuscript.

## Competing interests

D. P. and T. K. serve as consultants to Pixium Vision. D. P.'s, B. W.'s, and Z. C.'s patents related to retinal prostheses are owned by Stanford University and licensed to Pixium Vision, the details of which are disclosed below. All other authors declare no competing interests. Patent 1: - patent applicant: Stanford University - name of inventor(s): D. Palanker, A. Vankov, M. Blumenkranz - application number: US 7,047,080 - status of application: issued - specific aspect of manuscript covered in patent application: use of photovoltaic pixels for retinal prosthetics Patent 2: - patent applicant: Stanford University - name of inventor(s): Zhijie Chen, Daniel Palanker, Bingyi Wang - application number: PCT/US2022/020498 - status of application: in review - specific aspect of manuscript covered in patent application: current steering by optical control of the photovoltaic pixels.
