## [Peer Review File · Nature Communications]

REVIEWER COMMENTS

Reviewer #5 (Remarks to the Author):

The authors have provided responses to my previous comments which have alleviated some concerns. I have further comments in relation to this re-submission:

1. I accept that in this study the authors measured a prosthetic visual acuity with 20 μ m pixels that matched an acuity limit found with natural light in a group of healthy sighted rats that were also tested (both measurements about 26-27 μ m). However, it is clearly shown in their previous study (Ho, E., et al. Characteristics of prosthetic vision in rats with subretinal flat and pillar electrode arrays. *J Neural Eng* 16, 066027 (2019)), in Figure 4 that natural acuity limit in Long Evans rats is 17 μ m. So which natural acuity limit is correct? Could the authors explain this discrepancy between the two studies? If the 17 μ m natural acuity is indeed possible in rats, then the main claim of this paper should be toned down. Note, it is still remarkable that 26 μ m prosthetic acuity can be achieved and the authors should be commended for realising a device that can do so, but there is then no requirement to claim that it is simply a "limit of rats" that their device faces, implying that had rats not been used in this study, prosthetic acuity with 20 μ m pixels would be higher.
2. Thank you for providing the additional statistical results in the response. Could the authors please provide in Figure 7 raw traces of the 26 μ m prosthetic and natural light induced VEPs so that it is clear that both are clearly distinguishable from the background noise.
3. Figure 7 does not include the confidence intervals of the fit. Could the authors please also add this information.
4. Device safety: Thank you for including a statement about safety in the manuscript. 3mC/cm² for a 3 mW/mm² irradiance is quite a high charge density and is on the upper end of what SIROF electrodes can support for long term stimulation (limit \sim 2.4 mC/cm²). Therefore, in patients if this high irradiance is required, it won't be possible. The monopolar arrangement may bring down the requirement but it is

unknown. Therefore, the authors should briefly discuss the implications of using high irradiance values in patients with 20um pixel devices. Also, the Huang et al 2021 paper cited did not appear to test safety at this upper limit.

Reviewer #6 (Remarks to the Author):

The ability to use the same contact for active stimulation and passive return path without complex circuitry is an important advance. Passive discharge of electrode charge through a diode and unstimulated electrodes should be inherently safe by staying within polarization window before electrolysis can occur. Planar geometry should reduce foreign body reaction and facilitate reliability of SiC passivation demonstrated previously by this group (Xin et al., 2016, J. Neural Engng), but this needs discussion and citations.

Issues to be addressed:

It would be useful to contrast for the general reader the strategy of polarizing bipolar cells to modulate their transmitter release onto retinal ganglion cells versus the previously commercialized epiretinal prostheses that use brief (<1 ms) pulses with high currents (>100 uA), which likely directly activate action potentials in retinal ganglion cells and, unfortunately, their proximal axons.

The electrical field orientation changes with the vertical distance from electrode contacts to bipolar cells and the spatial gradient of the stimulation pattern. The Discussion mentions subretinal debris in human degenerated retina but ignores the human foreign body response, which may produce thicker encapsulation and greater displacement than in rat. Both would substantially affect both the achievable visual acuity and the thresholds, particularly for detecting objects with high spatial frequencies. The authors' models should show how these effects vary with distance but the only simulations presented are for highly optimistic distances of 10 um (Fig. 3) and 20 um (Fig. 4).

There is no information about the materials and fabrication of the electrode array. Presumably there are some similarities to previous designs published by this group but that is never explicitly stated. In particular, what were the external surfaces, including the exposed electrodes (presumably SIROF but applied according to what protocol?) and intervening passivation layer (perhaps SiC as described by Xin et al., 2016, but uncited). These will be important factors in the foreign body reaction, whose thickness critically limits the achievable spatial resolution and power efficiency of the design.

There is no information about the timing of the experiments. How long after implantation were the electrophysiological measurements obtained? If obtained repeatedly, were there any changes over time? When were the OCT measurements of these structures obtained (supplementary material shows one image at 6 weeks) and how did they change over time? Was there any histological evaluation at necropsy?

Discussion assumes applicability of Fig. 1 cartoon to high resolution foveal vision in humans, but that cytoarchitecture is very different from that modeled and studied in the rat retina. Fig. 1 needs discussion of differences between rat and human peripheral and foveal retina organization and dimensions and the degenerative reorganization that extends to inner retina in various forms of blindness.

Figs. 3C&D and S1 need scale for horizontal and vertical dimensions.

We would like to thank the referees for the time and effort dedicated to providing feedbacks on our manuscript, and we appreciate the opportunity of revision and improvement. We addressed the comments point-by-point below (in blue) and made corresponding changes in the manuscript.

Reviewer #5 (Remarks to the Author):

The authors have provided responses to my previous comments which have alleviated some concerns. I have further comments in relation to this re-submission:

1. I accept that in this study the authors measured a prosthetic visual acuity with 20um pixels that matched an acuity limit found with natural light in a group of healthy sighted rats that were also tested (both measurements about 26-27um). However, it is clearly shown in their previous study (Ho, E., et al. Characteristics of prosthetic vision in rats with subretinal flat and pillar electrode arrays. J Neural Eng 16, 066027 (2019)), in Figure 4 that natural acuity limit in Long Evans rats is 17um. So which natural acuity limit is correct? Could the authors explain this discrepancy between the two studies? If the 17um natural acuity is indeed possible in rats, then the main claim of this paper should be toned down. Note, it is still remarkable that 26um prosthetic acuity can be achieved and the authors should be commended for realising a device that can do so, but there is then no requirement to claim that it is simply a "limit of rats" that their device faces, implying that had rats not been used in this study, prosthetic acuity with 20um pixels would be higher.

There is a range of values for rats' natural acuity in the literature. Our own measurements in the past with two different setups provided different results: $27 \pm 9 \mu\text{m}^1$ and $17 \pm 5 \mu\text{m}^2$, both of which are in the range of previously reported values. Since the amplitude of the VEP measurements and the noise level depend on the setup, we wanted to make sure that our comparison between prosthetic and natural vision are performed with the same electronics, same level of anesthesia, identical type of data analysis, etc. Therefore, we measured the natural acuity in LE rats with our current VEP setup. Our result, $27.9 \pm 2.8 \mu\text{m}$, corresponding to a mean of 1.15 cpd, closely matches our first measurement¹ and the majority of the acuity values in the literature: 1 to 1.6 cpd³, 1 to 1.18 cpd⁴, 0.5 to 4 cpd⁵, and 1.15 cpd⁶. Such an agreement gives us confidence in our reported value.

The discrepancy between Ho et al.² (17 μm or 1.88 cpd) and our current result could stem from the differences in the optical projection system (Ho et al. used a 532 nm laser, while we now used a more common broad-band white light), noise level of the electronics, fitting method, level of anesthesia, etc.

We now added a statement about motivation of the natural acuity measurements in Methods, and toned down our claim regarding potential acuity in patients.

2. Thank you for providing the additional statistical results in the response. Could the authors please provide in Figure 7 raw traces of the 26um prosthetic and natural light induced VEPs so that it is clear that both are clearly distinguishable from the background noise.

We now updated Figure 7A to include more waveforms, including the ones near the acuity limit (26 or 27 μm) for both, prosthetic and natural vision in rats. In both cases, the waveforms taken with gratings narrower than 26 μm are indistinguishable from the noise trace. Above the acuity limit, the VEP waveforms began to exhibit their characteristic shape, distinct from noise. For prosthetic vision, the signature peak around 20 ms latency begins to appear at 26 μm stripe width, and its magnitude

increases with wider gratings, maintaining at the same latency. For natural vision, the signature peak appears later due to photoreceptors, and its latency decreases with increasing stripe width.

3. Figure 7 does not include the confidence intervals of the fit. Could the authors please also add this information.

We now added shaded regions representing the 95% confidence interval of the fits in Figure 7B-D, in addition to the written acuity range. The supplemental figure S6 has also been updated to indicate the confidence intervals of the fit in linear scale.

4. Device safety: Thank you for including a statement about safety in the manuscript. $3\text{mC}/\text{cm}^2$ for a $3\text{mW}/\text{mm}^2$ irradiance is quite a high charge density and is on the upper end of what SIROF electrodes can support for long term stimulation (limit $\sim 2.4\text{mC}/\text{cm}^2$). Therefore, in patients if this high irradiance is required, it won't be possible. The monopolar arrangement may bring down the requirement but it is unknown. Therefore, the authors should briefly discuss the implications of using high irradiance values in patients with $20\mu\text{m}$ pixel devices. Also, the Huang et al 2021 paper cited did not appear to test safety at this upper limit.

Clinical trials of the PRIMA implants demonstrated their safety under $3.5\text{mW}/\text{mm}^2$ irradiance with 10 ms pulses at 30 Hz rep. rate^{7,8}. These bipolar pixels are composed of two diodes connected in series, generating the maximum voltage of about 1.2 V. Since our new monopolar design includes only one diode per pixel, the maximum voltage and the maximum charge density is twice lower than with the 2-diode pixels, and hence it can only be safer. We now added these considerations to the discussion of safety in the manuscript.

Reviewer #6 (Remarks to the Author):

The ability to use the same contact for active stimulation and passive return path without complex circuitry is an important advance. Passive discharge of electrode charge through a diode and unstimulated electrodes should be inherently safe by staying within polarization window before electrolysis can occur. Planar geometry should reduce foreign body reaction and facilitate reliability of SiC passivation demonstrated previously by this group (Xin et al., 2016, J. Neural Engng), but this needs discussion and citations.

In addition to two citations of our paper about fabrication of these implants with the detailed description of the passivation coating (SiC)⁹, we now added an explicit mention of this coating in Methods: "In this array, each pixel has a SIROF-coated active electrode of $18\mu\text{m}$ in diameter, exposed to electrolyte, and a photosensitive area insulated by SiO_2 and SiC".

Issues to be addressed:

It would be useful to contrast for the general reader the strategy of polarizing bipolar cells to modulate their transmitter release onto retinal ganglion cells versus the previously commercialized epiretinal prostheses that use brief ($<1\text{ms}$) pulses with high currents ($>100\mu\text{A}$), which likely directly activate action potentials in retinal ganglion cells and, unfortunately, their proximal axons.

Difference between the subretinal and epiretinal signal encoding and stimulation strategies are described in many review articles¹⁰⁻¹². Since here we focus on the issues of resolution with subretinal

stimulation, we did not include any discussion of the epiretinal approach (nor other alternatives). We now added a sentence to the Introduction about this difference:

“This strategy is very different from the direct stimulation of the ganglion cells by epiretinal electrodes, which requires proper encoding of the spiking patterns for each cell type to reproduce the retinal signal processing [18].”

The electrical field orientation changes with the vertical distance from electrode contacts to bipolar cells and the spatial gradient of the stimulation pattern. The Discussion mentions subretinal debris in human degenerated retina but ignores the human foreign body response, which may produce thicker encapsulation and greater displacement than in rat. Both would substantially affect both the achievable visual acuity and the thresholds, particularly for detecting objects with high spatial frequencies. The authors’ models should show how these effects vary with distance but the only simulations presented are for highly optimistic distances of 10 μm (Fig. 3) and 20 μm (Fig. 4).

Current paper is focused on modeling and measurements in rats. As to humans, the debris layer separating the INL from the implant in patients may be due to many reasons, including the potential foreign body response. Detailed modeling of the retinal stimulation in human patients is very challenging because it not only involves proper solving of the Poisson equation with thousands of non-linear pixels coupled via electrolyte for various illumination patterns, but also should relate to various thresholds observed in human patients. Since the first submission of this paper to Nature BME over a year ago, we performed such analysis, including various strategies for optical current steering, and recently uploaded it to bioRxiv¹³, as well as the analysis of the exact limits of the bipolar pixel size for human patients¹⁴. These papers are in review, and we now added references to these articles. We also toned down the expectations for human vision in Abstract, Discussion and in Conclusions.

There is no information about the materials and fabrication of the electrode array. Presumably there are some similarities to previous designs published by this group but that is never explicitly stated. In particular, what were the external surfaces, including the exposed electrodes (presumably SIROF but applied according to what protocol?) and intervening passivation layer (perhaps SiC as described by Xin et al., 2016, but uncited). These will be important factors in the foreign body reaction, whose thickness critically limits the achievable spatial resolution and power efficiency of the design.

We refer to our previous publication⁹ with a detailed description of the implants’ fabrication, coatings, and other specifications in two places: first in the Introduction: “Here, we present an alternative approach to high-resolution prosthetic vision with planar monopolar subretinal implants¹⁴.” And then in Methods: “We modeled the electric field in the retina generated by a subretinal 30 μm -thick array 1.5 mm in diameter with 40- μm photovoltaic pixels, described in detail in¹⁴.” We now expanded the latter sentence to make a more explicit mention of the insulating materials: “In this array, each pixel has a SIROF-coated active electrode of 18 μm in diameter, exposed to electrolyte, and the photosensitive area is insulated by SiO₂ and SiC.” The material of the implant (Si), its external insulation/passivation (SiO₂/SiC), the electrodes coating (SIROF), as well as the additional protection of the back and sides on the device by Ti (see Figure 6 in Huang et al., 2021⁹ for more details) are exactly the same as in the previous generation of the implants for rats and for humans.

As to the foreign body reaction, clinical results⁷ demonstrated that such implants are well-tolerated in subretinal space, and are separated from the INL by about 30-40 μm , which remains stable over time. These numbers are mentioned in the Discussion: “In addition, subretinal debris in AMD patients, separating the INL from the implant by about 30-40 μm ⁸ may impose additional constraints on attainable resolution, compared to rodents²⁹.”

There is no information about the timing of the experiments. How long after implantation were the electrophysiological measurements obtained? If obtained repeatedly, were there any changes over time?

When were the OCT measurements of these structures obtained (supplementary material shows one image at 6 weeks) and how did they change over time? Was there any histological evaluation at necropsy?

The electrophysiological measurements were conducted both acutely (within an hour after implantation) and chronically (over months later). We did not observe differences in the stimulation threshold or acuity limit at different times. OCT imaging was also performed repetitively and showed good integration of the implant with the retina with little difference over time. We plan to perform histological evaluation after the end of life of these animals. We now added a statement about timing in the Methods.

Discussion assumes applicability of Fig. 1 cartoon to high resolution foveal vision in humans, but that cytoarchitecture is very different from that modeled and studied in the rat retina. Fig. 1 needs discussion of differences between rat and human peripheral and foveal retina organization and dimensions and the degenerative reorganization that extends to inner retina in various forms of blindness.

The fact that human and rat retinas have different resolution limits is at the core of this paper. Here, we present that with 20 μm pixels in rats, acuity is limited by the natural resolution limit of rats. In human retina, where higher resolution is attainable, we expect the prosthetic visual acuity to exceed 20/100, corresponding to 1.2 times of pixel pitch. Clinical results have demonstrated that despite the potential rewiring of the degenerating retina in patients, visual acuity closely matched the 100 μm pixel pitch⁸. The fact that rewiring may limit the acuity with smaller pixels is acknowledged in the Discussion: "It is important to keep in mind however, that retinal rewiring in AMD patients⁶ may limit the fidelity of prosthetic vision."

Figure 1 is a schematic diagram illustrating where the photovoltaic pixels are placed and how they are oriented. In humans, the implants are placed in the fovea, but over its 2 mm width, the extent of convergence varies from a single photoreceptor per BC in the very center to a few tens of photoreceptors per BC couple millimeters away. In Figure 1, pixel width corresponds to approximately 5 photoreceptors in diameter, or 20 photoreceptors per pixel area.

Figs. 3C&D and S1 need scale for horizontal and vertical dimensions.

We added a scale bar to Figure 3C, corresponding to half of the 40 μm pitch of the electrodes. This image, as well as the Figure 3D, represent the electric field map, and hence the vertical and horizontal dimensions are the same. Similarly, we added a scale bar to Figure S1 in the updated supplemental materials.

References

1. Lorach, H. *et al.* Photovoltaic restoration of sight with high visual acuity. *Nat. Med.* **21**, 476–482 (2015).
2. Ho, E. *et al.* Characteristics of prosthetic vision in rats with subretinal flat and pillar electrode arrays. *J. Neural Eng.* **16**, 066027 (2019).
3. Seymoure, P. & Juraska, J. M. Vernier and grating acuity in adult hooded rats: The influence of sex. *Behav. Neurosci.* **111**, 792–800 (1997).

4. Silveira, L. C., Heywood, C. A. & Cowey, A. Contrast sensitivity and visual acuity of the pigmented rat determined electrophysiologically. *Vision Res.* **27**, 1719–1731 (1987).
5. Boyes, W. K. & Dyer, R. S. Pattern reversal visual evoked potentials in awake rats. *Brain Res. Bull.* **10**, 817–823 (1983).
6. Birch, D. & Jacobs, G. H. Spatial contrast sensitivity in albino and pigmented rats. *Vision Res.* **19**, 933–937 (1979).
7. Palanker, D., Le Mer, Y., Mohand-Said, S., Muqit, M. & Sahel, J. A. Photovoltaic Restoration of Central Vision in Atrophic Age-Related Macular Degeneration. *Ophthalmology* **127**, 1097–1104 (2020).
8. Palanker, D., Le Mer, Y., Mohand-Said, S. & Sahel, J. A. Simultaneous perception of prosthetic and natural vision in AMD patients. *Nat. Commun.* **13**, (2022).
9. Huang, T. W. *et al.* Vertical-junction photodiodes for smaller pixels in retinal prostheses. *J Neural Eng* **17** (2021).
10. Palanker, D. Electronic Retinal Prostheses. in *The Senses: A Comprehensive Reference* 642–668 (Elsevier, 2021). doi:10.1016/B978-0-12-809324-5.24123-0.
11. Palanker, D. Visual Prosthesis, Optoelectronic Devices. in *Encyclopedia of Computational Neuroscience* (eds. Jaeger, D. & Jung, R.) 1–4 (Springer, 2018). doi:10.1007/978-1-4614-7320-6_665-2.
12. Palanker, D. & Goetz, G. Restoring sight with retinal prostheses. *Phys. Today* **71**, 26–32 (2018).
13. Chen, Z. C. *et al.* Photovoltaic Implant Simulator Reveals the Resolution Limits in Subretinal Prosthesis. 2022.06.30.498210 Preprint at <https://doi.org/10.1101/2022.06.30.498210> (2022).
14. Wang, B.-Y., Chen, Z. C., Bhuckory, M., Goldstein, A. K. & Palanker, D. Pixel size limit of the PRIMA implants: from humans to rodents and back. 2022.06.29.498181 Preprint at <https://doi.org/10.1101/2022.06.29.498181> (2022).

REVIEWERS' COMMENTS

Reviewer #5 (Remarks to the Author):

Thank you for addressing the previous comments. I have only one more point given the latest response from the authors. Since the authors in regard to point#1 have explained that rats with healthy vision can have a large range of measured acuity (and as high as 4 cpd) that can depend on a range of factors, they should include this point in the discussion possibly somewhere near line 191. The revised text should include the statement(s) that acuity in natural vision can vary widely and therefore it is possible that the prosthetic acuity limit measured with the 20um pixels in this study may be due to other unknown factors and not necessarily an inherent limit of the retina.

Reviewer #6 (Remarks to the Author):

The Ms changes and rebuttal text are generally responsive to the comments of the referees. One important issue addressed in the rebuttal needs to be addressed in the Ms text. The whole point of this article is the relationship between the spatial resolution of the retinal circuitry itself vs. the attainable spatial resolution of the stimulation. This is described for the rat, which has a uniformly distributed retinal histology rather than a fovea. The technology is only of interest for clinical application to humans and the article discusses the resolution achieved by forerunners of the new design applied to human fovea. The location and orientation of the various cellular elements in the human fovea is complex, even more so following degenerative changes, so a definitive model is beyond the scope of this article. Nevertheless, the reader needs to be advised of these differences and their potential implications for achievable resolution. The programmability of the passive return electrodes may even provide a mechanism for coping with these complexities.

We would like to thank the referees for the time and effort dedicated to providing feedbacks on our manuscript. We addressed the comments point-by-point below (in blue) and made corresponding changes in the manuscript.

Reviewer #5 (Remarks to the Author):

Thank you for addressing the previous comments. I have only one more point given the latest response from the authors. Since the authors in regard to point#1 have explained that rats with healthy vision can have a large range of measured acuity (and as high as 4 cpd) that can depend on a range of factors, they should include this point in the discussion possibly somewhere near line 191. The revised text should include the statement(s) that acuity in natural vision can vary widely and therefore it is possible that the prosthetic acuity limit measured with the 20um pixels in this study may be due to other unknown factors and not necessarily an inherent limit of the retina.

We added a statement regarding the potentially other unknown limiting factors in prosthetic acuity to Discussion: "we demonstrated that prosthetic visual acuity with 20 μm pixels is not limited by the pixel pitch, but rather matches the 28 μm natural visual acuity limit in rats, indicating that with such small pixels, resolution appears to be limited by the retinal network integration rather than by the sensor size, although other unknown limiting factors cannot be excluded."

Reviewer #6 (Remarks to the Author):

The Ms changes and rebuttal text are generally responsive to the comments of the referees. One important issue addressed in the rebuttal needs to be addressed in the Ms text. The whole point of this article is the relationship between the spatial resolution of the retinal circuitry itself vs. the attainable spatial resolution of the stimulation. This is described for the rat, which has a uniformly distributed retinal histology rather than a fovea. The technology is only of interest for clinical application to humans and the article discusses the resolution achieved by forerunners of the new design applied to human fovea. The location and orientation of the various cellular elements in the human fovea is complex, even more so following degenerative changes, so a definitive model is beyond the scope of this article. Nevertheless, the reader needs to be advised of these differences and their potential implications for achievable resolution. The programmability of the passive return electrodes may even provide a mechanism for coping with these complexities.

We added to Discussion a statement about the acuity decreasing with eccentricity in human retina and its potential role in prosthetic vision: "Natural resolution becomes worse than 20 μm at eccentricity exceeding 7 degrees¹, or 2 mm from the fovea. Therefore, the

centrally placed implant smaller than 4 mm in diameter with 20 μm pixels is not expected to exceed the natural resolution in human retina.”

References

1. Westheimer, G. Visual Acuity. in *Adler's Physiology of the eye, Clinical Application* (eds. Moses, R.A. & Hart, W.M.) (The C. V. Mosby Company, St. Louis, 1987).